# Advancing Cognitive Load Detection in Simulated Driving Scenarios Through Deep Learning and fNIRS Data

**DOI:** 10.3390/s25164921

**Published:** 2025-08-09

**Authors:** Mehshan Ahmed Khan, Houshyar Asadi, Mohammad Reza Chalak Qazani, Ghazal Bargshady, Sam Oladazimi, Thuong Hoang, Ghazal Rahimzadeh, Zoran Najdovski, Lei Wei, Hirash Moradi, Saeid Nahavandi

**Affiliations:** 1Institute for Intelligent Systems Research and Innovation (IISRI), Deakin University, Geelong, VIC 3216, Australia; houshyar.asadi@deakin.edu.au (H.A.); thuong.hoang@deakin.edu.au (T.H.); grahimzadeh@deakin.edu.au (G.R.); zoran.najdovski@deakin.edu.au (Z.N.); lei.wei@deakin.edu.au (L.W.); 2College of Science and Engineering, James Cook University, Townsville, QLD 4814, Australia; mohamadreza.chalakqazani@jcu.edu.au; 3Faculty of Science and Technology, University of Canberra, Canberra, ACT 2617, Australia; ghazal.bargshady@canberra.edu.au; 4Hermann-Hesse Str. 17, 72221 Haiterbach, Germany; sam.ozi@lenus-healthcare.de; 5Building Controls GmbH, Schießmauer 18, 92283 Lauterhofen, Germany; hirash.moradi@b-c.bayern; 6Swinburne Research, Swinburne University of Technology, Hawthorn, Melbourne, VIC 3122, Australia; snahavandi@swin.edu.au

**Keywords:** cognitive load, fNIRS, driving simulator, EEGNet, deep learning model

## Abstract

The shift from manual to conditionally automated driving, supported by Advanced Driving Assistance Systems (ADASs), introduces challenges, particularly increased crash risks due to human factors like cognitive overload. Driving simulators provide a safe and controlled setting to study these human factors under complex conditions. This study leverages Functional Near-Infrared Spectroscopy (fNIRS) to dynamically assess cognitive load in a realistic driving simulator during a challenging night-time-rain scenario. Thirty-eight participants performed an auditory n-back task (0-, 1-, and 2-back) while driving, simulating multitasking demands. A sliding window approach was applied to the time-series fNIRS data to capture short-term fluctuations in brain activation. The data were analyzed using EEGNet, a deep learning model, with both overlapping and non-overlapping temporal segmentation strategies. Results revealed that classification performance is significantly influenced by the learning rate and windowing method. Notably, a learning rate of 0.001 yielded the highest performance, with 100% accuracy using overlapping windows and 97% accuracy with non-overlapping windows. These findings highlight the potential of combining fNIRS and deep learning for real-time cognitive load monitoring in simulated driving scenarios and demonstrate the importance of temporal modeling in physiological signal analysis.

## 1. Introduction

Driver inattention remains one of the primary contributing factors to road traffic accidents worldwide. A critical subset of inattention is driver distraction, which specifically refers to the diversion of attention from tasks that are essential for safe driving to non-driving-related activities [1,2]. These distractions can stem from both internal sources (e.g., using infotainment systems, texting, eating, or conversing with passengers) and external sources (e.g., looking at billboards or roadside events). Such distractions compromise the driver’s situational awareness, reaction time, and overall ability to maintain safe control of the vehicle [3]. Although the advancement of automated driving technologies is expected to significantly reduce road traffic accidents, potentially eliminating up to 90% of incidents caused by human error, driver monitoring remains crucial [4]. Even in partially automated vehicles, drivers are often required to maintain situational awareness and resume control under certain conditions. Hence, understanding and detecting driver cognitive workload and distraction is vital for the design of safer and more adaptive driver-assistance systems.

Despite these limitations, driving simulators remain the safest and most controlled method for exposing participants to challenging or hazardous driving scenarios without placing them at risk of physical harm, collisions, or property damage [5]. Simulators offer a highly customizable environment in which various factors such as traffic density, road geometry, weather conditions, time of day, and distraction-inducing elements can be manipulated to replicate real-world driving challenges. This enables researchers to systematically investigate human behavior and physiological responses under controlled but realistic conditions that would otherwise be unsafe or impractical to study on actual roads [6]. In this study, the elevated levels of cognitive workload observed during the experiments were intentionally induced through a combination of environmental stressors (e.g., night-time driving and heavy rainfall) and secondary task demands, such as the auditory n-back task. These stressors were designed to replicate real-world multitasking demands and increase mental effort, thereby simulating complex driving situations that require increased attention, decision-making, and working memory.

Previous neuroimaging studies utilizing techniques such as Functional Near-Infrared Spectroscopy (fNIRS) have extensively investigated the relationship between cognitive workload and brain activity, with a particular focus on task-induced activation in specific cortical regions [7,8]. These studies consistently show that increased cognitive demands often manipulated through working memory tasks lead to heightened activation in areas such as the prefrontal cortex and parietal lobe, which are key regions involved in attention, executive function, and memory processing [9]. fNIRS-based studies have frequently reported load-dependent increases in blood oxygenation (HbO2) levels in the frontal cortex as n increases, typically showing a pattern where 2-back tasks elicit a stronger activation than 1-back tasks, which in turn elicit more than 0-back tasks [10,11]. However, this relationship is not always linear. In several studies, a non-linear activation pattern has been observed, where frontal activation may plateau or even decrease at the highest task difficulty, contrary to the expected trend [12]. This has been attributed to task disengagement, where participants may mentally give up when the cognitive demands exceed their processing capacity. Another explanation is that cortical activation may saturate, reaching a physiological ceiling beyond which no further activation is possible, regardless of task difficulty.

To analyze the complex and often non-linear patterns that are present in physiological data, a wide range of machine learning (ML) and deep learning (DL) algorithms have been increasingly employed across various domains [13,14,15]. These data-driven approaches are capable of capturing subtle variations and high-dimensional relationships that may not be easily detectable through traditional statistical methods [16,17]. In recent years, the fNIRS research community has also begun to adopt and explore the potential of ML and DL techniques for tasks such as cognitive state classification, mental workload detection, and brain–computer interface development. Their ability to automatically learn informative features from raw or minimally preprocessed data has opened new avenues for more accurate and scalable modeling of brain activity patterns using fNIRS signals. Traditional ML techniques such as Support Vector Machines (SVMs), Linear Discriminant Analysis (LDA), k-Nearest Neighbors (k-NN), and Random Forests have been widely used for the classification of workload levels [18,19]. More recently, DL approaches including Convolutional Neural Networks (CNNs), Long Short-Term Memory (LSTM) networks, and Gated Recurrent Units (GRUs) [20,21] have gained traction due to their ability to automatically extract and model temporal and spatial features from high-dimensional fNIRS signals [21,22]. These models are particularly useful in capturing subtle, non-linear patterns that traditional models might miss, offering new opportunities for robust and real-time workload classification.

The primary objective of this study is to investigate the impact of cognitive load on drivers within a controlled, high-fidelity simulated driving environment, specifically under challenging conditions such as night-time driving and heavy rainfall. These environmental stressors were chosen to closely replicate real-world situations where drivers often experience heightened mental demands due to reduced visibility, increased vigilance, and complex decision-making requirements. Unlike many previous studies that have primarily focused on only two levels of cognitive workload, typically low and high, this research introduces a three-levels-of-workload paradigm using an auditory-modified n-back task (0-back, 1-back, and 2-back). Furthermore, while prior studies often relied on traditional statistical methods for classification, such techniques are typically limited in their generalizability and tend to be highly subject-specific, performing well only on individual-level data. In contrast, our study employs a DL-based approach, which is capable of learning complex, non-linear patterns from large-scale physiological datasets. This enables the model to generalize more effectively across a broader population of participants, offering greater potential for real-time, scalable cognitive load detection in real-world driving scenarios. This approach demonstrates how incremental increases in cognitive demand affect driver performance and brain activity. A key aim of this study is also to examine whether these increasing levels of cognitive workload are associated with corresponding changes in cerebral blood oxygenation, as measured by fNIRS. Specifically, we investigate whether higher mental effort correlates with elevated levels of HbO2 in the prefrontal cortex, an area known to be involved in working memory, attention, and executive function.

## 2. Materials and Methods

A total of 38 drivers participated in this study. To ensure consistency in cognitive performance across participants, a set of specific inclusion criteria was established. All participants were required to possess a valid driver’s license, confirming their eligibility and basic driving competence. Additionally, individuals with any known history of mental health disorders, neurological conditions, or physical impairments that could potentially affect cognitive functioning were excluded from the study. This exclusion criterion was implemented to minimize potential confounding variables and ensure that variations in cognitive workload could be attributed more reliably to the experimental manipulations rather than underlying medical conditions. As a result, only data from clinically healthy participants with no self-reported or documented cognitive or physical impairments were included in the final analysis, thereby improving the internal validity of the findings. The study was approved by the Deakin University Human Research Ethics Committee (Project ID: 2021-181).

### 2.1. Driving Simulator

For this study, we utilized a driving simulator setup designed to deliver a realistic driving experience. At the core of the system was Next Level Racing Motion Platform V3 (Next Level Racing, Australia), securely mounted on the Traction Plus Platform. This combination was specifically chosen to enhance motion feedback, allowing participants to physically perceive vehicle dynamics such as acceleration, braking, and road vibrations, thereby improving the overall realism of the simulation. The visual interface of the simulator comprised three large 32-inch Samsung monitors, arranged in a panoramic configuration to provide a wide field of view. This setup was used in simulating peripheral vision and increasing participants’ situational awareness, both of which are critical for realistic driving behavior. To further enrich the tactile experience, a Thrust master T300 steering wheel and pedal system was integrated into the simulator. This system offered accurate force feedback, allowing participants to experience real-time steering resistance, road texture, and vehicular control with high fidelity. A visual illustration of the complete simulator setup is presented in Figure 1.

To replicate the feel and functionality of a real vehicle, the simulator was configured to mirror the driving dynamics and interior layout of a Toyota Fortuner SUV. This vehicle model was selected to maintain consistency in participants’ perception of vehicle handling, the cabin environment, and spatial awareness. For the driving scenarios, we employed Euro Truck Simulator 2 (ETS2) beta version 1.47 software, recognized for its realistic driving physics and extensive environmental conditions. ETS2 was selected for its capability to emulate a wide range of driving environments, including highway cruising, urban traffic navigation, and varying weather scenarios such as rain and fog. These features allowed us to construct diverse and cognitively demanding driving tasks that were representative of real-world conditions.

### 2.2. Secondary Cognitive Task

To simulate realistic multitasking demands and elevate cognitive load during the dual-task driving condition, an auditory-modified n-back task was employed in this study. This task was specifically designed to engage working memory and executive function while participants were simultaneously involved in a dynamic driving scenario. The cognitive load manipulation combined elements of the traditional n-back paradigm with a digit-span task, resulting in three graded levels of difficulty: 0-back, 1-back, and 2-back. The 0-back condition served as the baseline and required minimal cognitive effort, as participants simply identified a pre-specified target digit. In contrast, the 1-back and 2-back conditions progressively increased the memory load, requiring participants to continuously monitor and compare the current digit to the one presented one or two steps earlier in the sequence, respectively. Among these, the 2-back task was designed to impose the highest cognitive demand, thereby enabling the assessment of participants’ ability to manage an increased mental workload while driving. Auditory stimuli consisted of randomly selected spoken digits ranging from 0 to 9, presented through the simulator’s speakers in a consistent male voice at fixed intervals of 3.5 s. Each spoken digit lasted approximately 0.2 to 0.3 s, depending on the specific digit, with the remaining time allocated for participant response. This modality was chosen to prevent visual distraction and allow seamless integration with the visual demands of the driving task. Participants were instructed to provide their responses using two buttons, red and green, strategically mounted on the steering wheel for easy access, as depicted in Figure 2. The green button indicated a match between the current digit and the target digit, while the red button indicated a non-match. This hands-on response method ensured minimal physical distraction from the driving task, while maintaining engagement with the cognitive task.

The auditory n-back tasks were developed and implemented using the Python PsychoPy library [23]. PsychoPy was also utilized to capture participant responses via the red and green buttons mounted on the steering wheel. To ensure a smooth and immersive dual-task experience, PsychoPy was configured to operate in parallel with the driving simulator. This integration was critical for maintaining the ecological validity of the experiment, as it allowed participants to remain fully engaged in the simulated driving environment while concurrently performing the cognitive task.

### 2.3. Functional Near-Infrared Spectroscopy (fNIRS)

The hemodynamic activity of the prefrontal cortex was recorded using a high-density fNIRS device, NIRSIT (OBELAB Inc., Seoul, South Korea). This wearable neuroimaging system is equipped with 24 light sources (laser diodes) and 32 photodetectors, operating at two near-infrared wavelengths, 780 nm and 850 nm, to measure changes in cerebral blood oxygenation. Data acquisition was performed at a sampling rate of 8.138 Hz, allowing for continuous monitoring of brain activity throughout the experimental sessions. The source-detector pairs were arranged to create a dense coverage of the prefrontal region, with a fixed inter-optode distance of 1.5 cm. The raw optical density signals collected from the fNIRS device were pre-processed using OBELAB’s built-in Digital Signal Processing toolkit. This toolkit applied noise reduction and signal correction algorithms to enhance signal integrity. Following pre-processing, changes in HbO2 and Deoxygenation (HbR) concentrations were computed using the Modified Beer–Lambert Law [24], a standard approach for quantifying hemodynamic responses based on light absorption properties in biological tissues.

## 3. Experimental Procedure

The study began with the collection of written informed consent from all participants prior to their formal enrolment. Following consent, participants attended a comprehensive briefing session. During this session, they received both verbal and written instructions detailing the study’s purpose, the structure of the experimental tasks, and relevant safety protocols. After the briefing, participants underwent the sensor fitting procedure. They were equipped with the necessary physiological and neuroimaging equipment, including a high-density fNIRS system to monitor cerebral hemodynamics. Next, participants completed a simulator familiarization phase, during which they spent several minutes interacting with the driving simulator. Before the experimental tasks began, a one-minute resting-state baseline was recorded while participants sat quietly in the simulator. During this phase, all lights in the simulator setup were turned off to create a dark environment, enabling physiological signals to stabilize and establishing an individualized baseline for detecting task-induced changes. The experimental phase involved participants performing a series of structured driving tasks under controlled yet cognitively demanding conditions. The simulated environment mimicked real-world challenges, including night-time driving at approximately 1:00 a.m. and heavy rainfall. These conditions were intended to increase visual and attentional demands, thereby simulating scenarios that elevate cognitive load and mental fatigue in actual driving situations.

While navigating the simulated driving environment, participants concurrently performed the auditory-modified n-back task at varying difficulty levels (0-back, 1-back, and 2-back). The steering wheel-mounted red and green buttons reinforced dual-task coordination. This dual-task paradigm was designed to assess cognitive workload by requiring the allocation of attention and working memory resources across both the primary (driving) and secondary (n-back) tasks. Throughout the experimental session, fNIRS was used to continuously monitor changes in cerebral oxygenation within the prefrontal cortex. Combined with measures of driving performance and task accuracy, the fNIRS data provided insights into how cognitive load and environmental complexity interacted to affect multitasking performance in safety-critical settings.

## 4. Research Methodology

fNIRS data is collected from the prefrontal cortex while participants engage in a driving simulator and perform auditory n-back tasks (0-back, 1-back, and 2-back) to induce varying levels of cognitive load. The raw signals undergo a pre-processing pipeline and are segmented into 10 s, 20 s, and 30 s windows using both overlapping and non-overlapping strategies, allowing for the analysis of temporal resolution and classification performance. These segments are then fed into EEGNet, a compact convolutional neural network originally designed for EEG data, which is adapted here to classify the cognitive workload based on hemodynamic patterns. An overview of the data processing and classification pipeline, including fNIRS acquisition, pre-processing, windowing strategies, and EEGNet classification, is illustrated in Figure 3.

### 4.1. Data Pre-Processing

First, to normalize the feature scales and mitigate the influence of varying baseline values across channels, the fNIRS signals were standardized using the Standard Scaler method as shown in Equation (1). This transformation was applied independently to each channel across the dataset.(1)$$x\prime =\frac{x-ú}{õ}$$

In the above equation, x is the original feature value, ú is the mean, and õ is the standard deviation computed across the training dataset. This ensures that the data for each channel has zero mean and unit variance, which improves the convergence of learning algorithms and helps in comparing features on the same scale.

After standardization, the fNIRS time-series data was segmented into temporal windows to extract meaningful features that reflect short-term variations in cognitive load. This segmentation is a critical step in time-series analysis, particularly for physiological signals such as fNIRS, where brain activation patterns fluctuate over time. In this study, two types of segmentation strategies were adopted: overlapping windows and non-overlapping windows. The overlapping window method utilizes a sliding window approach, in which each new segment shares a portion of its data with the previous segment. This technique enhances the temporal resolution of the dataset, potentially capturing transient or transitional changes in cognitive states more effectively. In contrast, the non-overlapping window strategy divides the entire signal into consecutive, discrete segments with no repetition. This method reduces computational load and redundancy in the data but may risk missing subtle transitions between cognitive states.

To determine the most effective temporal resolution for classifying cognitive load from fNIRS signals, we systematically experimented with three different time window lengths: 10 s, 20 s, and 30 s. These window durations were selected based on their frequent use in prior neuroimaging and physiological signal processing studies [16,25], where they have been shown to provide a meaningful balance between capturing sufficient temporal context and maintaining model responsiveness. The window sizes reflect a balance between capturing fine-grained temporal dynamics and ensuring sufficient context for robust model training. Furthermore, these window lengths align well with the typical temporal profile of the hemodynamic response function in fNIRS data, which typically peaks around 6–8 s post-stimulus, making shorter windows potentially inadequate and much longer windows unnecessary.

### 4.2. EEGNet Model

To classify cognitive load across three levels (0-back, 1-back, and 2-back), this study employed the EEGNet model architecture [26], originally developed for EEG-based brain–computer interface applications. EEGNet was chosen for its compact design and proven ability to decode neurophysiological signals with relatively few parameters. The structure of the EEGNet model is shown in Table 1. The model was trained using the Adam optimizer and was optimized with categorical cross-entropy loss, which is suitable for multi-class classification tasks. Training was conducted over 200 epochs to enable the model to learn discriminative spatiotemporal features from the input fNIRS data. The EEGNet architecture consists of three sequential blocks designed to progressively extract and integrate spatial and temporal features from the input signals.

#### 4.2.1. Block 1: Spatial Feature Extraction

This block begins with an input layer followed by two key convolutional operations. A 2D convolution is first applied to extract low-level features across time and channels. This is followed by a depthwise convolution, which applies a separate filter to each channel individually. Unlike standard convolutions, this method significantly reduces the number of trainable parameters while still capturing meaningful spatial patterns. Batch normalization follows each convolutional operation, standardizing feature distributions and facilitating stable learning. The depthwise convolution specifically enhances training efficiency and mitigates overfitting, which is particularly valuable when working with relatively small neuroimaging datasets.

#### 4.2.2. Block 2: Separable Convolution for Spatiotemporal Integration

Block 2 employs a separable convolution, which decouples the learning of temporal and spatial patterns. It starts with a depthwise convolution that independently processes each feature map over time, enabling the model to capture temporal dynamics within each channel. This is followed by a pointwise convolution (1 × 1), which combines the temporally filtered signals across channels, allowing the model to learn cross-channel dependencies. This two-stage approach reduces computational complexity while preserving the ability to represent brain activity patterns distributed across time and spatial locations. By explicitly separating the modeling of temporal and spatial structures, the model becomes better suited to detecting subtle changes in cognitive workload.

#### 4.2.3. Block 3: Classification and Output

The final block of the EEGNet model maps the learned high-level features to class predictions. It begins with a flattening layer that transforms the multi-dimensional feature maps into a one-dimensional vector suitable for classification. This vector is then passed through a dense layer that projects the features onto three output nodes, each representing one of the cognitive load levels (0-back, 1-back, and 2-back). A SoftMax activation follows, generating a probability distribution over the three classes, which enables the model to produce confidence scores for each prediction. The model is trained using the categorical cross-entropy loss function, which compares the predicted probabilities to the true class labels.

### 4.3. Validation Framework

Cross-validation is a fundamental strategy in ML for assessing a model’s performance, robustness, and generalizability to unseen data [27,28]. Among the various techniques available, k-fold cross-validation is widely adopted due to its balance between computational efficiency and reliability [29]. In k-fold cross-validation, the dataset is partitioned into k equal-sized subsets or “folds.” The model is trained on k−1 folds and tested on the remaining one. This process is repeated k times, each time with a different fold used as the test set. The results across all folds are then averaged to obtain a more generalized estimate of model performance.

In this study, k was set to 5, meaning the dataset was divided into five equal parts. During each of the five iterations, a different subset was used for validation while the remaining four subsets were used for training. This ensures that each sample in the dataset is used exactly once for testing, thereby minimizing bias and variance in the performance evaluation. The final accuracy, as well as other performance metrics, was reported as the mean of the five folds, offering a comprehensive view of the model’s classification capability. To explore the effect of temporal granularity on model performance, the dataset was processed using three distinct window lengths: 10 s, 20 s, and 30 s. For each window size, we generated both overlapping and non-overlapping segments from the time-series fNIRS data. After segmentation, the data was randomly shuffled to remove any sequence bias before being divided into five folds. This randomized shuffling step ensures that samples from the same time period or task condition do not disproportionately influence any single fold. For each window size configuration (e.g., 10 s windows), the same fold partitioning was used to train the model under three different learning rates (LRs): 0.1, 0.01, and 0.001. This consistent fold usage across LRs allows for a fair comparison of how LR influences classification performance under identical data conditions.

### 4.4. EEGNet Hyperparamter Tuning

To ensure optimal classification performance of cognitive workload using fNIRS signals, the EEGNet model was fine-tuned through extensive hyperparameter tuning. EEGNet, originally developed for EEG-based brain–computer interface applications, was adapted in this study to classify hemodynamic responses captured by fNIRS. Due to the distinct temporal resolution and signal characteristics of fNIRS, model hyperparameters had to be carefully customized for different input segment durations.

We experimented with three temporal resolutions commonly cited in the literature for cognitive load analysis: 10 s, 20 s, and 30 s. At a sampling rate of 8.138 Hz, these correspond to 81, 163, and 244 samples per window, respectively. As the input window length increases, the amount of temporal information grows, requiring deeper convolutional processing to effectively extract relevant temporal and spatial features. As shown in Table 2, several model hyperparameters were dynamically adjusted for each window size. For instance, the temporal kernel size (Kernel 1) in the first convolutional layer was increased progressively from 32 to 128, while the depthwise convolution kernel (Kernel 2), which captures longer-term dependencies, was scaled from 8 (for 10 s windows) to 32 (for 30 s windows). Likewise, the number of temporal filters (F1) and the number of pointwise filters (F2) were doubled at each step to match the temporal complexity of the longer windows. The depth multiplier (D) for spatial filtering was held constant at 2 across all configurations, maintaining a balance between model complexity and training stability.

These adjustments ensured that EEGNet could handle varying levels of temporal resolutions without underfitting or overfitting to shorter or longer signal segments. In addition to the window-specific parameters, several hyperparameters were kept uniform across all training runs to ensure consistency and fair evaluation, as illustrated in Table 3. These included architectural and training-level configurations such as the type of dropout, pooling strategy, activation function, and loss function. All models used Mean Pooling, ELU activation, and Cross Entropy Loss as the classification objective. The final convolutional kernel size was determined automatically based on the input window size to preserve model integrity and allow seamless adaptation.

The Adam optimizer was used in all configurations for its ability to adaptively adjust LRs during training. The batch size of 64 was selected to balance computational efficiency with convergence stability. The full EEGNet training pipeline is summarized in Algorithm 1, which outlines a structured and repeatable algorithm for training and validating the model. For each window size (10 s, 20 s, 30 s), the time-series fNIRS data was segmented into overlapping and non-overlapping windows, from which the top 50 features were selected using ANOVA-based feature selection. The dataset was then normalized, randomly shuffled, and split using 5-fold cross-validation. To evaluate the influence of the LR on classification performance, three LRs were tested, 0.1, 0.01, and 0.001, for each window size. Importantly, the same folds were used across LR experiments for each window duration to ensure comparability.
**Algorithm 1:** EEGNet training with ANOVA feature selection and varying temporal windows.
**Input:**
Labelled fNIRS dataset D,
Window sizes W={10 s,20 s,30 s},
Sampling frequency $f_{s}=8.138\;Hz$,
Learning rates LR ={0.1, 0.01, 0.001},
Number of top ANOVA features K = 50,
Number of folds F = 5,
Number of epochs E**Output:**
Trained EEGNet models F for each configuration**1:**
**for** each window size w∈W
**do****2:**

$n\to w\times f_{s}$          //Convert seconds to number of samples**3:**

(X,y)← Extract segments of length n from D**4:**

$X^{\prime }\leftarrow$ Select top K features from X using ANOVA**5:**

 D← Normalize and prepare dataset $\left(X^{\prime },y\right)$**6:**

**for** each learning rate η∈LR **do****7:**


**for** fold = 1 to F **do****8:**



$D_{train},D_{test}\leftarrow$ Split D using fold F**9:**



Initialize EEGNet F with window size n and dynamic parameters $\left(F_{1},F_{2},\;kernel\;length\right)$**10:**



Initialize optimizer with LR η**11:**



**for** epoch = 1 to E **do****12:**




Train F on $D_{train}$ using Cross Entropy Loss**13:**




Evaluate F on $D_{test}$ and record metrics**14:**



**end** for**15:**



Save model F and results for (w,η, fold)**16:**


**end for****17:**

**end for****18:**
**end for****19:**
**return** all trained EEGNet models F

This tuning and training procedure allowed us to systematically investigate the effect of temporal window size and LR on classification accuracy, AUC, precision, recall, and F1-score across different experimental configurations.

## 5. Results and Discussions

Given that the dataset comprises 204 fNIRS channels, a feature selection step was necessary to reduce dimensionality and identify the most informative signals associated with cognitive load during simulated driving. To achieve this, we employed the Analysis of Variance (ANOVA) method, a statistical approach commonly used in neuroimaging to evaluate the significance of each feature with respect to class separation. Figure 4 presents the top 50 ranked features identified through ANOVA-based feature selection. As illustrated in the figure, a greater proportion of the selected features correspond to HbO2 channels rather than HbR. This observation suggests that changes in blood HbO2 are more strongly associated with variations in cognitive load than changes in HbR.

This finding aligns with prior research indicating that HbO2 signals tend to exhibit higher sensitivity to task-related neural activation, particularly within the prefrontal cortex, where cognitive processes such as working memory and attention are regulated s [10,30,31]. The predominance of HbO2 features among the top-ranked channels implies that blood oxygenation dynamics are more robust indicators of cognitive workload in dual-task conditions, such as driving while performing a secondary cognitive task (e.g., the n-back task).

After selecting the top 50 features from the fNIRS dataset based on their relevance to cognitive load classification, further refinement was performed through correlation analysis. This step aimed to assess the degree of linear dependency between features to ensure that each selected feature contributes uniquely to the model, without redundancy. Highly correlated features can introduce multicollinearity, which may degrade the performance and interpretability of ML models by overemphasizing certain aspects of the signal while masking others.

To address this, a pairwise correlation matrix was computed across the selected features using Pearson’s correlation coefficient. This matrix quantifies the linear relationship between feature pairs, with values ranging from −1 (perfect negative correlation) to +1 (perfect positive correlation). The resulting correlation structure is visualized in Figure 5, which presents a heatmap of the correlation coefficients. This visualization highlights the relationships among features and provides insight into the overall redundancy or complementarity of the selected features. By ensuring a low degree of correlation among the input variables, the final dataset becomes more robust and informative for subsequent model training, reducing overfitting and improving generalization.

To train and evaluate the EEGNet model, we employed a 5-fold cross-validation strategy. This approach was chosen to ensure robust model performance and to minimize the potential for overfitting, particularly given the limited size and high dimensionality of the dataset. In each fold, the dataset was partitioned into five equal subsets: four were used for training, and one was used for validation. This process was repeated five times, with each subset serving as the validation set once, allowing for a more generalized assessment of model performance.

For model optimization, we utilized the Adam optimizer, known for its adaptive LR capabilities and efficiency in training deep neural networks. To examine the effect of LR on model performance, three different LRs were tested: 0.1, 0.01, and 0.001. The model was trained over 200 epochs with a batch size of 63, which was selected to balance convergence stability and computational efficiency. In addition to model training, we also explored the impact of temporal segmentation strategies on classification performance. Specifically, we evaluated the model using both overlapping and non-overlapping time window segments of 10 s, 20 s, and 30 s. This segmentation was applied to the time-series input data to investigate how different windowing strategies affect the model’s ability to capture temporal patterns associated with cognitive load. The results for the overlapping segmentation evaluations are presented in Table 4, while the results for the non-overlapping segmentation are detailed in Table 5.

The evaluation results based on overlapping window segments reveal that a window size of 30 s, in combination with an LR of 0.001, yields the highest classification performance, achieving an accuracy of 100%. This suggests that longer overlapping windows allow the EEGNet model to capture more stable and comprehensive patterns of neural activity associated with cognitive load, leading to near-perfect model performance. The overlapping approach benefits from the redundancy introduced by the windowing technique, which may help in smoothing out transient noise and enhancing the temporal context for the model. In contrast, for the non-overlapping segmentation, the highest accuracy of 97% is also achieved using an LR of 0.001, but in this case, the best performance is observed with a shorter window size of 10 s. Unlike the overlapping scenario, increasing the window size in the non-overlapping setup does not necessarily improve performance. This may be due to the fact that longer non-overlapping segments could introduce greater variability between segments or result in fewer training samples, thereby reducing the model’s generalizability. These findings demonstrate that while the LR of 0.001 consistently yields the best performance across both segmentation strategies, the optimal window size appears to be context dependent. Specifically, overlapping windows benefit more from longer durations, likely due to richer temporal information and redundancy, whereas non-overlapping windows may perform better at shorter durations, which offer more training samples and better granularity. Therefore, the highest classification performance is not solely determined by window size but rather by the interaction between the segmentation strategy and the temporal resolution used in pre-processing.

In addition to the segmentation results, an in-depth analysis of LR effects reveals distinct optimization dynamics between overlapping and non-overlapping segmentation strategies, as illustrated in Figure 6a,b. For overlapping segments, model performance varies dramatically across different LRs, ranging from approximately 56–59% accuracy at an LR of 0.1 to near-perfect performance (99.95–100%) at an LR = 0.001. This substantial variance suggests that overlapping segmentation introduces a more complex optimization landscape, likely due to the presence of redundant information across temporally adjacent windows. High LRs in this context may result in unstable convergence due to conflicting gradient updates from correlated segments. Conversely, when a low LR is applied (LR = 0.001), the model benefits from greater training stability and is better able to extract meaningful temporal patterns from the overlapping windows. In contrast, non-overlapping segments show more stable and gradual performance improvements across LRs, with accuracy ranging from 65–70% at LR = 0.1 to 85–97% at LR = 0.001. This relatively narrow performance band suggests that non-overlapping segmentation yields a more straightforward optimization process, as the absence of segment redundancy reduces the complexity of gradient updates. The model remains relatively robust even under aggressive LRs, but it exhibits slightly lower peak performance compared to the overlapping setup. This may be attributed to the limited temporal context available in non-overlapping windows, which constrains the model’s ability to learn cross-segment dependencies.

Further analysis of the performance difference between segmentation strategies, summarized in Figure 7, highlights an LR-dependent interaction that challenges typical assumptions about the advantages of data augmentation through overlapping. At higher LRs (LR = 0.1), non-overlapping segmentation outperforms overlapping by 9–11%, indicating that redundant data may hinder learning when combined with rapid weight updates. At a moderate LR (LR = 0.01), the performance gap narrows to 2–5%, suggesting a balance between training stability and data richness. However, at a low LR (LR = 0.001), overlapping windows outperform non-overlapping by a margin of 12–15%, demonstrating their advantage when training conditions allow for stable convergence.

These findings emphasize that neither segmentation strategy is universally superior; rather, their effectiveness depends critically on the LR and overall training configuration. While overlapping segments offer rich temporal information, they require careful tuning to avoid optimization instability. In contrast, non-overlapping segments provide robustness under a wider range of hyperparameter settings but may limit peak performance. Therefore, the interaction between the segmentation strategy and LR must be carefully considered to achieve optimal model generalization and efficiency.

## 6. Limitations and Future Directions

Despite the promising results, this study presents several limitations that warrant consideration and offer important directions for future research. First, while the use of a high-fidelity driving simulator provides a controlled and safe testing environment, it cannot fully replicate the dynamic, unpredictable conditions of real-world driving. This limitation may affect the ecological validity of the findings and restrict the model’s applicability in naturalistic driving contexts. Second, the participant sample (N = 38), although sufficient for preliminary analysis, may not fully reflect the demographic and cognitive diversity of the broader driving population. Factors such as age, driving experience, and individual cognitive profiles can significantly influence physiological responses and, consequently, model generalizability. Future studies should aim to recruit more demographically and cognitively diverse participant cohorts including variations in age, driving experience, and cognitive capacity to improve the robustness and generalizability of workload detection models across different user populations. In addition, the integration of more advanced driving simulators featuring high-fidelity motion-cueing algorithms could significantly enhance the ecological validity of experimental setups [32,33]. These systems are better equipped to replicate complex sensory and vestibular feedback associated with real-world driving, including acceleration, braking, and lateral movement, which are critical for eliciting authentic cognitive and physiological responses [34,35]. By combining participant diversity with realistic simulation environments, future research can build more scalable and context-aware models that are suitable for deployment in real-world driver monitoring and assistance systems.

Third, the experimental design employed a random-split 5-fold cross-validation approach. While this method is widely accepted, applying it to time-series physiological data such as fNIRS data may lead to data leakage [36], where temporally adjacent or similar patterns from the same subject appear in both training and validation sets. This can artificially inflate model performance and limit its ability to generalize [37]. Although a random split was used due to the relatively small sample size, future research should adopt subject-wise or session-wise cross-validation to more realistically evaluate generalizability, particularly in deployment scenarios targeting individual-independent performance. Another limitation relates to the scope of data acquisition. This study focused solely on fNIRS signals from the prefrontal cortex; a region associated with executive function. However, cognitive workload is influenced by distributed neural processes that involve multiple brain regions. To address this, eye-tracking data were collected using Pupil Core glasses and ECG data were recorded using an Equivital system as backup modalities. While these were not analyzed in the current study, they offer valuable opportunities for future work. Expanding cortical coverage or integrating complementary data sources such as driving behavior, ECG, and eye-tracking may improve model robustness and enable the detection of cognitive states through multimodal data fusion.

Additionally, although the auditory n-back task effectively modulated cognitive workload, it does not encompass the full spectrum of challenges encountered during real-world driving such as emotional stress, unexpected obstacles, verbal interaction, or decision-making under time constraints. Incorporating more ecologically valid secondary tasks could improve the realism and relevance of workload models. The DL model used EEGNet also introduces interpretability challenges. Like many neural networks, EEGNet functions as a black-box model, making it difficult to understand the basis of its predictions. This limitation is particularly critical in safety-sensitive applications such as driver monitoring. Future research should explore Explainable AI (XAI) techniques, including saliency mapping, SHAP values, and layer-wise relevance propagation, to increase model transparency and support trust and accountability in decision-making.

Finally, while EEGNet achieved near-perfect classification under certain configurations, especially with overlapping window strategies, such high accuracy on a relatively small and well-controlled dataset raises concerns about overfitting. Redundant information in overlapping segments may inflate performance, particularly in combination with random validation splits. Future research should rigorously test model performance using independent datasets, varied driving contexts, and real-time scenarios to confirm generalizability. To further enhance real-world applicability, future work may also investigate lightweight DL architectures and transfer learning strategies that enable model adaptation across users. These advancements could support scalable, real-time cognitive workload monitoring for deployment in intelligent driver-assistance systems and other human–machine interaction domains.

## 7. Conclusions

To systematically investigate varying levels of cognitive workload, we developed a dual-task experimental paradigm in which participants were required to perform a primary driving task concurrently with a secondary cognitive task, a modified version of the n-back task delivered through the auditory modality. This setup was carefully designed to replicate the cognitive demands encountered in real-world multitasking situations, such as managing complex navigation while processing verbal information. The n-back task was implemented at three distinct levels of difficulty: 0-back (representing a low cognitive workload), 1-back (moderate workload), and 2-back (high workload). By systematically varying the task difficulty, we aimed to elicit clearly differentiated cognitive states that could be measured through physiological responses during the driving simulation.

This approach allows us to evaluate the sensitivity and effectiveness of fNIRS in detecting subtle changes in cognitive load under realistic conditions. While previous studies have primarily focused on two-level workload comparisons (e.g., low vs. high), our three-tiered workload design introduces a more granular framework for understanding how cognitive demand escalates across multiple task intensities, offering richer insights into the brain’s adaptive response to increasing mental strain. In addition to the experimental design, we employed the EEGNet DL architecture to analyze the recorded fNIRS signals. EEGNet, originally developed for EEG data classification, has been adapted in our study to process and classify hemodynamic responses captured by fNIRS, enabling automated workload detection. A key methodological contribution of this work is the systematic evaluation of both overlapping and non-overlapping temporal window segmentation strategies, which remain underexplored in the existing literature. We also examined the impact of LR on model performance, finding that a low LR of 0.001 consistently achieved the highest classification accuracy across conditions. Our results indicate that longer overlapping windows (30 s) yield superior performance due to the enriched temporal context and redundancy, while shorter non-overlapping windows (10 s) perform better likely because of reduced variability and increased training samples. These findings highlight the importance of segmentation and optimization strategies to the specific characteristics of the data and model architecture. This study demonstrates the potential of fNIRS-based DL models for real-time cognitive workload detection in complex task environments. Future work will extend these findings by incorporating multimodal signals and subject-wise validation to further improve generalizability and ecological validity.

## Figures and Tables

**Figure 1 sensors-25-04921-f001:**
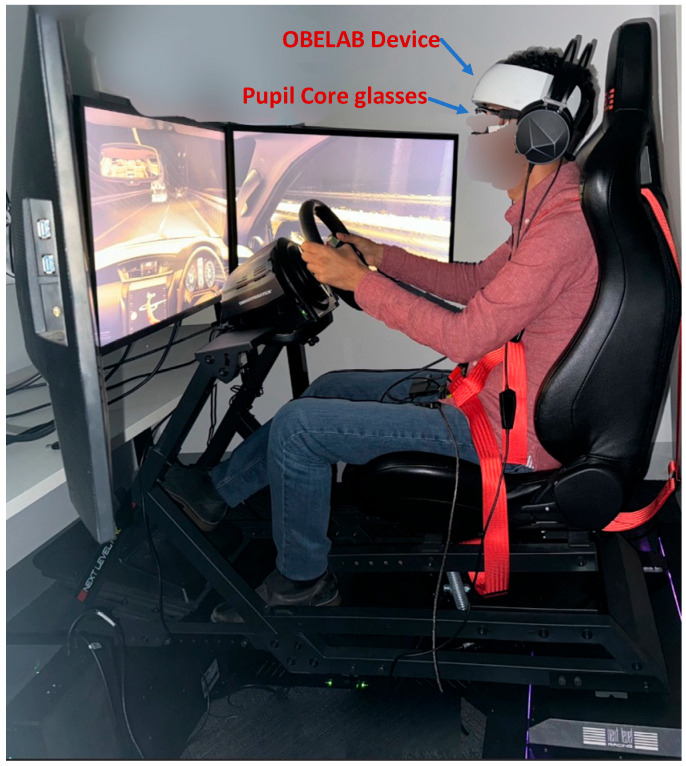
Driving simulator setup designed to replicate real-world vehicle dynamics and driving conditions using motion platforms and responsive control hardware.

**Figure 2 sensors-25-04921-f002:**
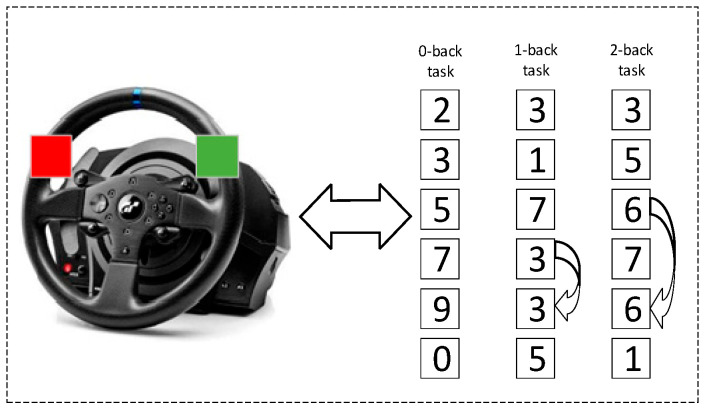
Steering wheel interface used during the auditory modified n-back task.

**Figure 3 sensors-25-04921-f003:**
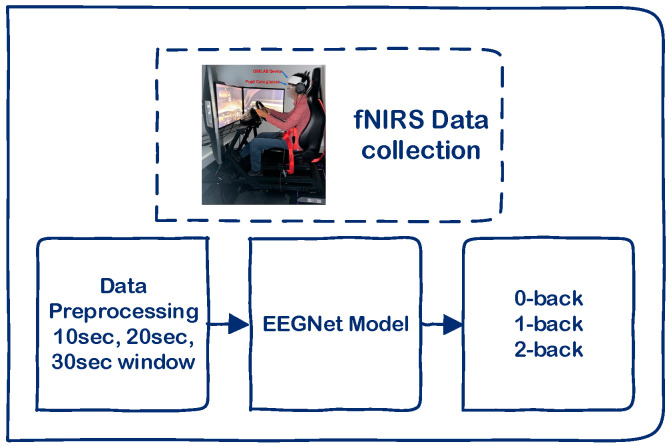
Schematic representation of the fNIRS-based cognitive workload classification pipeline.

**Figure 4 sensors-25-04921-f004:**
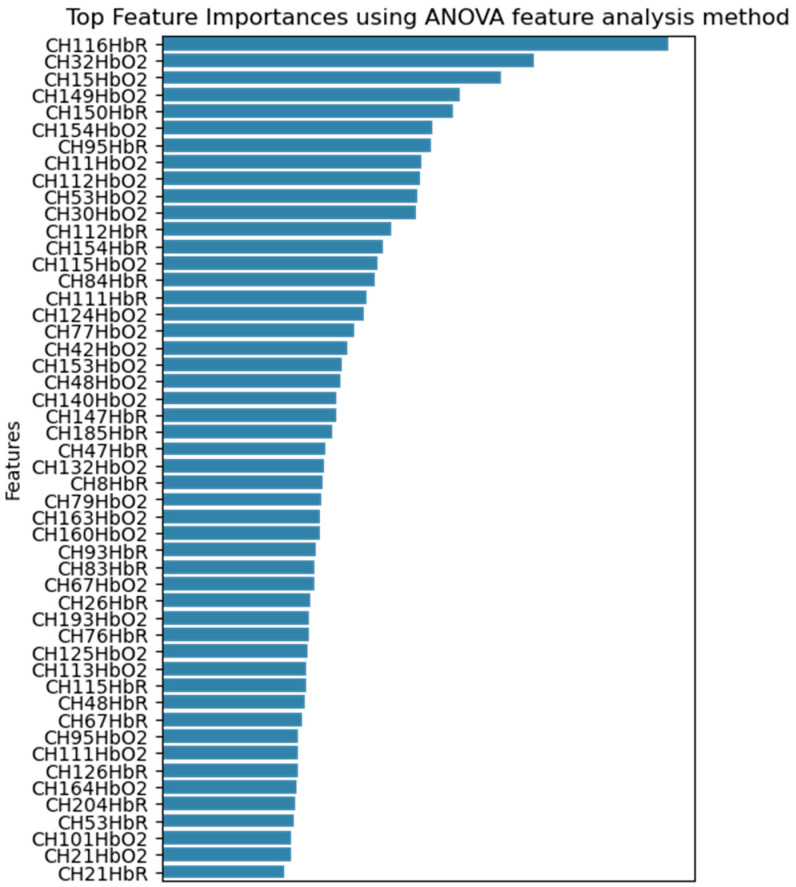
Visualization of the top 50 most significant fNIRS channels identified through ANOVA analysis.

**Figure 5 sensors-25-04921-f005:**
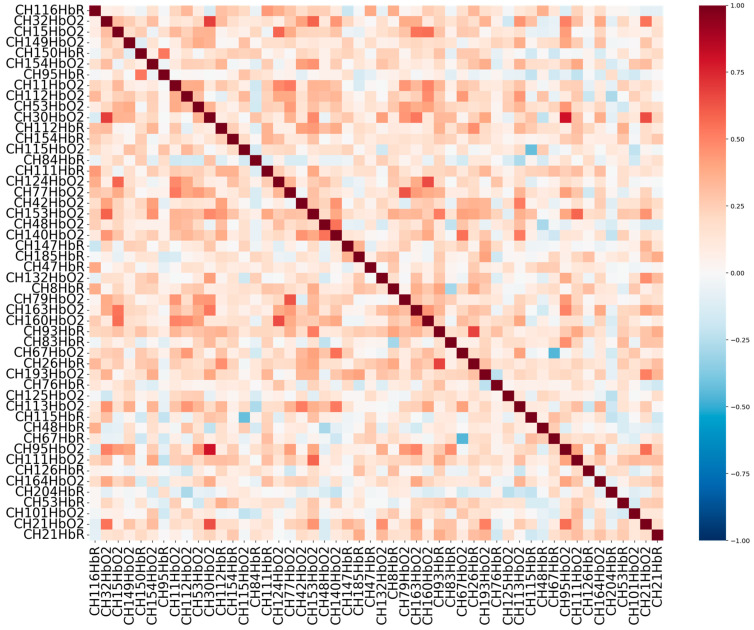
Correlation heatmap of the top 50 selected fNIRS features, illustrating the pairwise relationships among features after feature selection.

**Figure 6 sensors-25-04921-f006:**
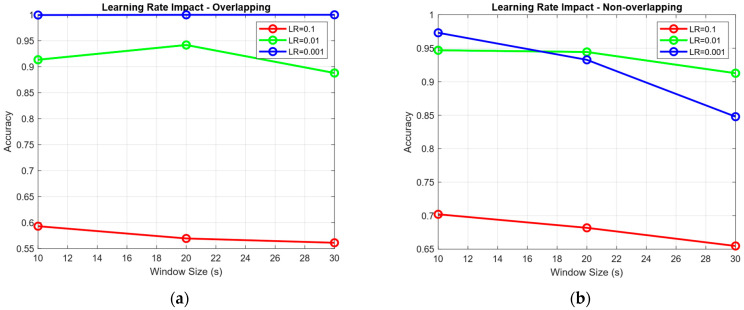
(**a**) Impact of LR on overlapping time window performance. A sharp accuracy gain at LR = 0.001 demonstrates that stability is critical when training with overlapping fNIRS segments. (**b**) Non-overlapping segmentation performance across varying LRs. The model shows stable convergence and lower peak accuracy.

**Figure 7 sensors-25-04921-f007:**
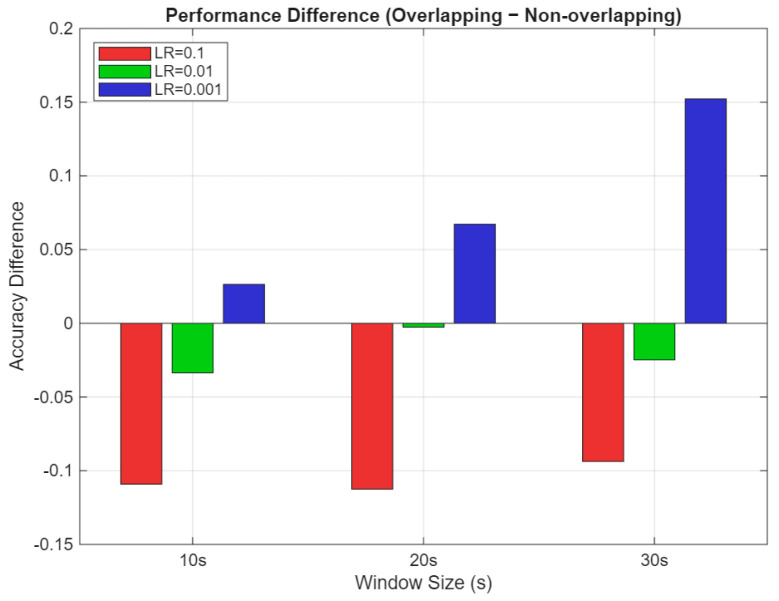
Performance difference between overlapping and non-overlapping segmentation strategies across LRs.

**Table 1 sensors-25-04921-t001:** Layer-wise configuration of the EEGNet model, illustrating the sequence of operations used to extract discriminative spatiotemporal features from fNIRS input for multi-class classification.

Type	Parameters	Output Shape
Conv2D	Input channels = 1, Output channels = F1, Kernel size = (1, kernel length), Padding = (0, Kernel length//2), Bias = False	[Batch size, F1, Number of channels, Number of time samples]
BatchNorm2D	Number of features = F1	[Batch size, F1, Number of channels, Number of time samples]
Conv2DWithConstraint	Input channels = F1, Output channels = F1·D, Kernel size = (Number of channels, 1), Maximum norm = 1, Bias = False	[Batch size, F1·D, 1, Number of time samples]
BatchNorm2D	Number of features = F1·D	[Batch size, F1·D, 1, Number of time samples]
ELU Activation	-	[Batch size, F1·D, 1, Number of time samples]
AvgPool2D or MaxPool2D	Kernel size = (1, 4), Stride = (1, 4)	[Batch size, F1·D, 1, Number of time samples/4]
Dropout	*p* = drop probability	[Batch size, F1·D, 1, Number of time samples/4]
Conv2D (Depthwise)	Input channels = F1·D, Output channels = F1·D, Kernel size = (1, 16), groups = F1·D	[Batch size, F1·D, 1, Number of time samples/4]
	Padding = (0, 8), Bias = False	
Conv2D (Pointwise)	Input channels = F1·D, Output channels = F2, Kernel size = (1, 1), Padding = (0, 0), Bias = False	[Batch size, F2, 1, Number of time samples/4]
BatchNorm2D	Number of features = F2	[Batch size, F2, 1, Number of time samples/4]
ELU Activation	-	[Batch size, F2, 1, Number of time samples/4]
AvgPool2D or MaxPool2D	Kernel size = (1, 8), Stride = (1, 8)	[Batch size, F2, 1, Number of time samples/32]
Dropout	*p* = drop prob	[Batch size, F2, 1, Number of time samples/32]
Conv2D	Input channels = F2, Output channels = N (classes), Kernel = (1, Final conv length), Bias = True	[Batch size, N, 1, 1]
Log Softmax	Dimension = 1	[Batch size, N, 1, 1]
Expression (squeeze)	-	[Batch size, N]

**Table 2 sensors-25-04921-t002:** Dynamic EEGNet hyperparameters adjusted per window length.

Parameter	Description	10 s	20 s	30 s
**Input Window Samples**	Number of samples per window (at 8.138 Hz)	81	163	244
**Dropout Rate**	Dropout fraction	0.25	0.25	0.25
**Kernel 1**	Length of temporal convolution in first Conv2D layer	32	64	128
**Kernel 2**	Depthwise conv kernel size (temporal dimension in Conv2D)	8	16	32
**F1**	Number of temporal filters to learn	8	16	32
**D**	Depth multiplier for spatial filtering	2	2	2
**F2**	Number of pointwise filters to learn (F1 × D)	16	32	64

**Table 3 sensors-25-04921-t003:** Fixed training parameters across all EEGNet experiments.

Parameters	Value
**Dropout type used**	Dropout
**Pooling type**	Mean Pooling
**Activation function**	ELU
**Final convolution kernel length (automatically computed from data shape)**	Auto
**Samples per batch during training**	64
**Optimization algorithm**	Adam
**Loss function**	CrossEntropyLoss
**Weight initialization method**	Xavier Uniform

**Table 4 sensors-25-04921-t004:** Classification performance of EEGNet using overlapping time window segments (10 s, 20 s, 30 s) across 5-fold cross-validation. Results are reported for various LRs, illustrating the impact of window size and overlap on model accuracy.

Window Size	Learning Rate	Accuracy	AUC	Recall	Precision	F1-Score
**10 s**	0.1	0.5929 ± 0.0491	0.7773 ± 0.0296	0.5929 ± 0.0491	0.6102 ± 0.0509	0.6102 ± 0.0509
**20 s**	0.1	0.5693 ± 0.0343	0.7737 ± 0.0348	0.5693 ± 0.0343	0.6345 ± 0.0260	0.6345 ± 0.0260
**30 s**	0.1	0.5610 ± 0.0267	0.7568 ± 0.0258	0.5610 ± 0.0267	0.5900 ± 0.0117	0.5900 ± 0.0117
**10 s**	0.01	0.9134 ± 0.0102	0.9651 ± 0.0064	0.9134 ± 0.0102	0.9138 ± 0.0106	0.9138 ± 0.0106
**20 s**	0.01	0.9418 ± 0.0157	0.9772 ± 0.0060	0.9418 ± 0.0157	0.9430 ± 0.0151	0.9430 ± 0.0151
**30 s**	0.01	0.8879 ± 0.0427	0.9485 ± 0.0179	0.8879 ± 0.0427	0.8932 ± 0.0384	0.8932 ± 0.0384
**10 s**	0.001	0.9995 ± 0.0002	0.9997 ± 0.0002	0.9995 ± 0.0002	0.9995 ± 0.0002	0.9995 ± 0.0002
**20 s**	0.001	0.9999 ± 0.0001	1.0000 ± 0.0000	0.9999 ± 0.0001	0.9999 ± 0.0001	0.9999 ± 0.0001
**30 s**	**0.001**	**1.0000 ± 0.0000**	**1.0000 ± 0.0000**	**1.0000 ± 0.0000**	**1.0000 ± 0.0000**	**1.0000 ± 0.0000**

**Table 5 sensors-25-04921-t005:** Classification performance of EEGNet using non-overlapping time window segments (10 s, 20 s, 30 s) across 5-fold cross-validation.

Window Size	Learning Rate	Accuracy	AUC	Recall	Precision	F1-Score
**10 s**	0.1	0.7020 ± 0.0219	0.8559 ± 0.0237	0.7020 ± 0.0219	0.7100 ± 0.0267	0.7100 ± 0.0267
**20 s**	0.1	0.6818 ± 0.0137	0.8233 ± 0.0331	0.6818 ± 0.0137	0.6838 ± 0.0199	0.6838 ± 0.0199
**30 s**	0.1	0.6547 ± 0.0257	0.7987 ± 0.0424	0.6547 ± 0.0257	0.6882 ± 0.0418	0.6882 ± 0.0418
**10 s**	0.01	0.9470 ± 0.0126	0.9759 ± 0.0077	0.9470 ± 0.0126	0.9486 ± 0.0117	0.9486 ± 0.0117
**20 s**	0.01	0.9444 ± 0.0081	0.9705 ± 0.0102	0.9444 ± 0.0081	0.9454 ± 0.0081	0.9454 ± 0.0081
**30 s**	0.01	0.9127 ± 0.0242	0.9378 ± 0.0427	0.9127 ± 0.0242	0.9179 ± 0.0245	0.9179 ± 0.0245
**10 s**	**0.001**	**0.9731 ± 0.0021**	**0.9888 ± 0.0011**	**0.9731 ± 0.0021**	**0.9732 ± 0.0020**	**0.9732 ± 0.0020**
**20 s**	0.001	0.9327 ± 0.0115	0.9678 ± 0.0036	0.9327 ± 0.0115	0.9346 ± 0.0106	0.9346 ± 0.0106
**30 s**	0.001	0.8478 ± 0.0190	0.9014 ± 0.0376	0.8478 ± 0.0190	0.8550 ± 0.0188	0.8550 ± 0.0188

## Data Availability

The datasets presented in this article are not publicly available due to privacy and ethical restrictions and will not be shared.

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
