# Peer review of "Advancing Cognitive Load Detection in Simulated Driving Scenarios Through Deep Learning and fNIRS Data"

_sensors, 2025, doi:10.3390/s25164921_

Round 1
Reviewer 1 Report
Comments and Suggestions for Authors
- The authors should significantly improve the presentation quality of the paper. Figure and table labels are inconsistently formatted, often duplicated, or broken, making it difficult to follow the content. This level of formatting issue is unacceptable for a submission and must be thoroughly addressed.
- The colorbar of the correlation heatmap should be expanded from -1 to 1, with white indicating 0.
- The authors rely solely on 5-fold cross-validation without using a separate test set. While cross-validation is useful for internal validation, it does not provide a reliable estimate of generalization performance. A held-out test set is essential to evaluate model performance properly, and its absence should be addressed or justified more clearly.
- “The evaluation results based on overlapping window segments reveal that a window size of 30 seconds, in combination with a learning rate of 0.001, yields the highest classification performance, achieving an accuracy of 100%. This suggests that longer overlapping windows allow the EEGNet model to capture more stable and comprehensive patterns of neural activity associated with cognitive load, leading to near-perfect model performance.” I do not agree with the conclusion that the 100% accuracy reflects the model's superior ability to capture cognitive load patterns. Since overlapping window segmentation inherently introduces significant redundancy between training and validation samples, the reported performance is likely overestimated due to a lack of data independence. The authors should critically reassess the validity of this result and consider using non-overlapping or held-out data to obtain a more realistic evaluation.
< !-- notionvc: 3eac6311-9a2c-40b1-b445-e622330f5ebe -->
Reviewer 2 Report
Comments and Suggestions for Authors
This study uses functional near-infrared spectroscopy (fNIRS) to assess the cognitive load while using a driving simulator. Participants performed an auditory n-back task while driving in a simulated rainy night. EEGNet was used to analyze the fNIRS data with overlapping and non-overlapping temporal windows. A learning rate of 0.001 yielded the highest performance, with 100% accuracy using overlapping windows and 97% accuracy with non-overlapping windows.
The authors should consider the following comments/suggestions to improve the overall quality of the manuscript:
1. The study involved humans. However, the manuscript lacks an ethics statement.
2. The figure numbers and table numbers are not sequentially numbered.
3. Define acronyms at the first instance of usage in the manuscript.
4. Lines 111-113: "Additionally, individuals with any known history of mental health disorders, neurological conditions, or physical impairments that could potentially affect cognitive functioning were excluded from the study." Could you please clarify who carried out this task and how it was done?
5. Line 132: "Error! Reference source not found..". Fixing the figure numbers should be able fix this error.
6. Figure 1: What was the purpose of the 'Pupil Core glasses'?
7. Were the tests conducted under dark conditions? If so, mention that in the manuscript.
8. Sections 2.1 and 2.2: Provide the company name and country information for the equipment and software used to benefit the readers.
9. Lines 158-160: "Auditory stimuli consisted of randomly selected spoken digits ranging from 0 to 9, delivered through the simulator’s speakers in a consistent male voice at fixed time intervals." What was the fixed time interval used?
10. Lines 183-185: "The source-detector pairs were arranged to create a dense coverage of the prefrontal region, with a fixed inter optode distance of 1.5 cm, as illustrated in 0". This sentence is incomplete and missing some information.
11. Section 3: Why was this section written in future tense?
12. Lines 233 and 273: "Error! Reference source not found..".
13. Lines 317-318: "0presents the top 50 ranked features identified through ANOVA-based feature selection" Error in the statement citing the figure.
14. Lines 325-327: "This finding aligns with prior research indicating that HbO2 signals tend to exhibit higher sensitivity to task-related neural activation, particularly within the prefrontal cortex, where cognitive processes such as working memory and attention are regulated." Cite the reference.
15. What was the rationale for choosing the window sizes of 10s, 20s, and 30s?
Comments on the Quality of English LanguageThe entire manuscript needs to be revised for minor English grammar and style errors. For example, Lines 259-260: "We used the varying window sizes was to explore the trade-off between temporal sensitivity and contextual information."
Reviewer 3 Report
Comments and Suggestions for Authors
The work presented by the authors is very interesting. During the review of the document, some questions arose that I would like to be addressed to clarify the authors' contribution.
The authors mention that they use fNIRS data records. However, the monitoring stage is not considered during the process, or is not clearly mentioned in the document. This stage allows for initial identification of each user's characteristics in order to measure changes in each of the parameters considered.
In Chapter 4, in section 4.2, the authors mention the algorithms and techniques used; however, there is little information related to their configuration and description of the procedure, which could be expanded to better understand the algorithms used.
In Chapter 5, the results section, the authors present two interesting figures. However, the success information for the proposed algorithm is not clearly visible in these images (it appears to be 100% successful for all parameters of the confusion matrix in Figure 2).
Figure 3, which is mentioned in line 341, does not exist.
Table 1 does not show significant information about the proposed recognition algorithm. Tests were performed at 0.1, 0.01, and 0.001, but it is not explained whether the same data was used for each analysis, that is, whether the algorithm's training set and the analysis set were changed during validation, and what percentage of the original record set was used. Finally, the missing references on lines 273, 233, 132, and 163 should be reviewed.
Round 2
Reviewer 1 Report
Comments and Suggestions for Authors
The authors have addressed the reviewer's comment. But the authors should proofread the whole manuscript before final submission to correct some typos such as in line 417-418. "are regulated s [10, 417 30, 31].. "
Reviewer 2 Report
Comments and Suggestions for Authors
The revised manuscript incorporates responses to all suggestions and comments. The modifications made and the details included have improved the overall quality of the manuscript.
Reviewer 3 Report
Comments and Suggestions for Authors During this review, the comments generated during the first review were analyzed, and the authors answered questions and revised the tables and algorithms presented in the document.